# Active Aging in ASEAN Countries: Influences from Age-Friendly Environments, Lifestyles, and Socio-Demographic Factors

**DOI:** 10.3390/ijerph18168290

**Published:** 2021-08-05

**Authors:** Sariyamon Tiraphat, Vijj Kasemsup, Doungjai Buntup, Murallitharan Munisamy, Thang Huu Nguyen, Aung Hpone Myint

**Affiliations:** 1ASEAN Institute for Health Development, Mahidol University, Salaya, Nakhon Pathom 73170, Thailand; vijj.kas@mahidol.ac.th (V.K.); doungjai.bun@mahidol.ac.th (D.B.); 2Faculty of Medicine Ramathibodi Hospital, Mahidol University, Bangkok 10400, Thailand; 3National Cancer Society of Malaysia, Kuala Lumpur 50300, Malaysia; murallimd@gmail.com; 4School for Preventive Medicine and Public Health, Hanoi Medical University, Hanoi 100000, Vietnam; nguyenhuuthang@hmu.edu.vn; 5Community Partners International (CPI) Bahan Township, Yangon 11201, Myanmar; aunghponemyint88@gmail.com

**Keywords:** age-friendly environment, active aging, healthy aging, healthy lifestyle, ASEAN, elderly population

## Abstract

Active aging is a challenging issue to promote older population health; still, there is little clarity on research investigating the determinants of active aging in developing countries. Therefore, this research aimed to examine the factors associated with the active aging of the older populations in ASEAN’s low and middle-income countries by focusing on Malaysia, Myanmar, Vietnam, and Thailand. The study is a cross-sectional quantitative research study using multi-stage cluster sampling to randomize the sample. The sample consists of 2031 older people aged 55 years and over, including 510 Thai, 537 Malaysian, 487 Myanmar, and 497 Vietnamese. We collected a quantitative questionnaire of age-friendly environmental scale and active aging scale based on the World Health Organization (WHO) concept. The predictors of active aging include age-friendly environments, lifestyles, and socioeconomic factors; the data are analyzed by using multiple logistic regression. After adjusting for other factors, we found that older people living in a community with higher levels of age-friendly environments are 5.52 times more active than those in lower levels of age-friendly environments. Moreover, the older population with healthy lifestyles such as good dietary intake and high physical activity will be 4.93 times more active than those with unhealthy lifestyles. Additionally, older adults with partners, higher education, and aged between 55 and 64 years will be 1.70, 2.61, and 1.63 times more active than those with separate/divorce/widow, primary education, and age at 75 years or higher, respectively. Our results contribute considerable evidence for ASEAN policy-making to promote active aging in this region.

## 1. Introduction

Globally, people live longer and the proportion of older people is growing, especially those in low-income and middle-income countries [1]. Southeastern Asia is where the rate of an aging population is overgrowing, with a growing rate of people aged 60 or above increasing from 9.8%, 13.7%, and 20.3% in 2017, 2030, and 2050, respectively [2]. Although life expectancy at birth in Southeast Asia has increased, older persons spend up to 10 years with impairments [3]. Various health problems that older people are facing include non-communicable diseases (NCDs), mental health, dementia, injuries, and disabilities due to declining functional ability [2]. In 1967, five leaders—the Foreign Ministers of Indonesia, Malaysia, the Philippines, Singapore, and Thailand—agreed to establish the Association of Southeast Asian Nations (ASEAN) in order to increase the quality of life for the Southeast Asian population. Currently, it includes the ten members of the Association of Southeast Asian Nations such as Brunei, Cambodia, Indonesia, Laos, Malaysia, Myanmar, the Philippines, Singapore, Thailand, and Vietnam. In order to prepare for rapidly aging societies, the members endorsed a “statement on active aging” to promote the older population’s quality of life and well-being in ASEAN in 2016 [4].

In the late 1990s, the World Health Organization (WHO) aimed to enhance the worldwide quality of life as people age and adopted the term “active aging” in order to optimize opportunities for health, participation, and security throughout individuals’ life courses. The word “active aging” allows people to realize the aging population with potential for physical, social, and mental well-being and to be ready to participate in society according to their needs with adequate protection, security, and care [5]. Active aging depends on various influences or determinants, such as personal, social, physical, health and social services, and economic conditions [5,6,7]. In order to maintain active aging, the WHO promoted the concept of creating an “age-friendly environment” that is accessible, equitable, inclusive, safe, and supportive environments optimize opportunities for health, participation, and security to enhance the quality of life as they age eventually. Eight domains of the age-friendly environment are associated with active aging based on WHO guidelines and they include urban spaces and built environment, housing, transportation, social participation, civic participation and employment, respect and social inclusion, access to community support and health services, and communication and information [8,9].

Many studies support the positive influence of age-friendly environments on active aging. For example, a previous study from the United States indicated that the age-supportive economic environment is critical for older health. The researchers found that the older population with involuntary job loss and prolonged unemployment negatively impacted financial well-being and resulted in poor emotional well-being and poor physical well-being [10]. Another U.S. research supported that unintentional job loss negatively affects the mental health of the older population and re-employment can improve the mental health status of involuntary job loss sufferers and retirees [11]. Another interestingly supportive environment is neighborhood public transportation; it was clear that public transportation accessibility influences active participation in daily activities [12]. For example, a previous study from Italy [13] confirmed that older Italians rely on cars. The researchers found that approximately half of older Italian people still make a daily trip by car and this was equal compared to other age groups. Other evidence from Canada showed that older Canadian people who drive are more than twice as likely to participate in out-of-home and social activities than non-drivers [14]. Therefore, securing mobility for the older population, such as possessing a driving license [15,16,17] or improving transport service [17], is significant for out-of-home activity participation in old age. 

In addition to a physical environment, a previous review [18] suggested that a social environment with respect and social inclusion may positively influence the older population’s physical and psychological well-being, whether in high and upper-middle-income countries or lower-middle-income countries [19]. The older individuals in such a supportive social environments could strengthen social relationships, improve self-confidence and self-esteem, feel valued, reduce social isolation, and become more physically active [18]. Previous research from Hong Kong also confirmed that among the domains of perceived neighborhood environment, “transportation” and “respect and social inclusion” were the physical and the social-environmental factors most strongly associated with self-rated health mediated by a sense of community [20]. 

In addition to a supportive environment, behavioral determinants, including a healthy diet and participation in physical activity, are also linked with active aging [21]. Some studies supported a positive association between the adherence to a healthy diet such as Mediterranean food (characterized by high intakes of fruit and vegetables, legumes, grains and cereals, fish and seafood, and nuts), muscle strength [22], and bone health in aging population [23]. In addition to diet, many studies research the benefit of physical activity for older people; a previous study specified that exercising the mind through intellectual stimulation, inquiry, and curiosity can sharpen older cognitive abilities [21,24]. In East Asia, age-related exercises focusing on martial arts such as karate or cognicise: Multicomponent exercise incorporating physical, cognitive, and social activities can enhance cognitive functioning by improving attention, resilience, and motor reaction time among older populations [25,26]. In Western countries, sports participation is a current standard policy to help older people maintain their independence and improve their general quality of life [27].

A purposeful goal of the ASEAN Foundation regarding the aging population is to promote the quality of life and well-being of the older people in ASEAN Nations. However, research evidence for understanding the determinants of active aging among older adults in ASEAN is quite limited. Therefore, this study aimed to examine the significant predictors of active aging among older adults in ASEAN. Active aging in this study comprises the three key pillars of active aging suggested by the World Health Organization, including self-reliance and independence, participation, and maintaining life security. In addition, the active aging determinants included age-friendly environments, behavioral determinants, and socio-demographic factors. Evidence of significant predictors of active aging will ensure the health and social authorities increase efforts for promoting active aging among ASEAN. 

## 2. Methodology

### 2.1. Description of Survey and Study Population

This research design was a cross-sectional interview household survey conducted for three months from January to March 2019. We used a multi-stage stratified sampling procedure and collected data in four metropolitans: Malaysia, Myanmar, Vietnam, and Thailand. First, we calculated the sample size using the older population aged 55 years and older in each metropolitan (Bangkok city, Hanoi city, Yangon city, and Kula Lumpur city), which is above 100,000 cases per city. Calculated by Taro-Yamane with 95% confidence, the sample size in each city was about 400 cases. After adding a missing rate of twenty-five percent, the final sample size for each city was approximately 500 cases. For all countries, the first stage was the selection of the city. In the second step, we randomly select three or four districts from the city. Thirdly, we chose each sub-district per district. At the final stage, every person 55 years of age and older living in the randomly selected households in the study area was eligible for the study. Finally, we randomly selected one individual for the face-to-face interview from all the eligible respondents in a household. The response rate in each country was 100%. The study population after excluding the observations with missing data was 2031 persons aged 55 years and older. The observations remained in the sample, including 537 from Malaysia, 497 from Vietnam, 487 from Myanmar, and 510 from Thailand. This research project received ethical approval from the “Research Ethics Committee of the Faculty of Social Sciences and Humanities, Mahidol University” (Certificate of Approval No. 2018/218.1809). Informed consent was obtained from all study participants.

### 2.2. Measures

#### 2.2.1. Perceived Age-Friendly Environments

All items of the perceived age-friendly environments scale were based on the “World Health Organization” guidebook: *Measuring the Age-Friendliness of Cities: A Guide to Using Core Indicator* [28]. The final questionnaire of the perceived age-friendly environments was adapted from the age-friendly environment questionnaire and validated with an older adult population in ASEAN [29]. It comprises seven items, including neighborhood walkability; neighborhood accessible public spaces and building; neighborhood respect and social inclusion; neighborhood job opportunity; neighborhood health and services information; neighborhood priority parking spaces; and neighborhood accessible public transportation stop, showing a good validity of Cronbach alpha 0.81 for this study. Each age-friendly environment is scored from 0 to 4 on a response ordinal scale (not at all, a little, moderately, mostly, and extremely). We summed the score from each item to obtain a total age-friendly environment and the scores ranged from 0–28. We then categorized three levels with low age-friendly environment (0–7), medium age-friendly environment (8–15), and high age-friendly environment (16–28).

#### 2.2.2. Perceived Active Aging

Perceived active aging in this study, adapted from the psychometric active aging scale for Thai adults [30], consisted of 26 items in three dimensions (Cronbach alpha = 0.93). It includes (A) self-reliance (eight items, e.g., “Each day, I try to do plenty of activities.”) (Cronbach alpha = 0.90); (B) participation (eight items, e.g., “I like to work for society without any concerns about getting paid.”) (Cronbach alpha = 0.86); (C) managing life security (ten items, e.g., “I can learn using new information technologies and facilitated equipment.”) (Cronbach alpha = 0.87). The response options ranged from 1 (not at all true) to 4 (very true). By summing the score from each subscale, the scores ranged from 28–103. A higher score referred to a more active aging performance and a lower score referred to low performance of active aging. Finally, we summed the scores of each subscale and dichotomized them as low and high, using the mean as the cut point. The cut point for total self-reliance, participation, and security are 68, 25, 18, and 25, respectively.

#### 2.2.3. Lifestyle Behaviors

For behavioral determinants, the scale consisted of five items (Cronbach alpha = 0.80) regarding dietary behaviors (e.g., “I try to select healthy foods”) and physical activity (e.g., “I regularly exercise at least three times a week”). The score ranges from 1 to 4 on a response ordinal scale (not at all true, slightly true, somewhat true, very true). Finally, we summed the scores of total healthy behaviors and dichotomized them as low and high, using the mean as the cut point = 14. 

#### 2.2.4. Socio-Demographic Variables

Socio-demographic variables included: age level (1 = 55–64 years; 2 = 65–74 years; 3 = 75 years and higher), gender (1 = male; 2 = female), educational level (1 = at least primary education (under 6 years of education), 2 = secondary education (6 years–9 years), 3 = higher education (more than 9 years of education), and marital status (1 = single; 2 = married/coupled; 3 = separate/divorce/widow), and country of residents (1 = Malaysia; 2 = Vietnam; 3 = Myanmar; 4 = Thai).

### 2.3. Data Analysis

Data were analyzed using Statistics Program for Social Sciences (SPSS) for windows, Version 21.0, Armonk, NY, USA: IBM Corp. Firstly, we conducted descriptive analysis to describe the sample. Next, in order to examine differences in the proportion of perceived active aging by socio-demographic, behavioral, and age-friendly conditions, we investigated them with Chi-square. Later, we performed multiple logistic regression to examine the predictors of active aging, including age level, gender, educational level, marital status, country of residents, health behaviors, and perceived total age-friendly environments. Finally, we placed all seven items of age-friendly environments including (1) walkability; (2)accessible public spaces and building; (3) respect and social inclusion; (4) job opportunity; (5) health and services information; (6) priority parking spaces; (7) accessible public transportation stop, controlling for health behaviors and socio-demographic factors, in multiple logistic regression model to investigate the significant predictors. We checked all predictors for collinearity, which was not concerning (Variance Inflation Factor = VIF ranging from 1.06–1.97). Statistical analyses were two-sided and significance was set at *p* < 0.05.

## 3. Results

### 3.1. Sample Characteristics

Participants in the total study sample (*N* = 2031) were at least 55 years old, less than two-thirds of the participants were female (63%), and less than half (45%) had completed elementary school. Distribution of the sample by age showed that almost half of the participants (47%) were between 55 and 64 years old, the rest were (37%) between 65 and 74 years old, and 16% were older than 75 years old. Most were married or coupled (60.6%), followed by separated/divorced/widowed (31.6%) and were single without partners (only 7.8%). Of the older population, 537 live in Malaysia, 497 in Vietnam, 487 in Myanmar, and 510 in Thailand. We found that about 45% of the participants reported high levels for healthy lifestyles, combining physical activity and a healthy diet. In contrast, about 56% of the participants reported low levels for healthy lifestyles (see Table 1).

### 3.2. Descriptive Statistics of Age-Friendly Environments and Active Aging among ASEAN Older Populations

Approximately 40% of the older adults perceived their neighborhood walkability as low, whereas about 30% perceived their neighborhood as medium and high. Regarding neighborhood public spaces and buildings, about 50% of the older adults perceived the neighborhood as low, whereas 30% and 25% perceived the neighborhood as medium and high. Approximately 25% of the older adults perceived low-level attitude towards respect and social inclusion, whereas about 30% and 43.6% perceived it as medium and high. Almost 70% of the older population perceived neighborhood job support as low, whereas only 16.8 and 16.4 perceived it medium and high. For neighborhood health and services information, about 40% of them perceived low level, whereas about 30% and 33.2 perceived medium and high. For neighborhood accessible bus stations, the older population perceived it as low, which was about 54%, whereas they perceived it as medium and high at 23%. Concerning the neighborhood elderly parking lot, the older population perceived it as low, about 55%, whereas they perceived it as medium and high for 23.2% and 21.4%. For the total age-friendly environment, approximately 28.0% of the older populations perceived it as low level, whereas about 46.6% and 26% perceived it as medium and high level (see Table 2).

Regarding active self-reliance, approximately 55% of the older population perceived high levels of active self-reliance. Regarding participation, about 42.0% of the older adults perceived high levels. In comparison, about 49% perceived that they had high levels of security. Approximately 50.4% of the older population perceived low levels for total active aging, whereas almost 50% of them perceived high levels (see Table 3).

### 3.3. Factors Associated with Active Aging

Chi-square tests show that significant factors were associated with active aging among ASEAN older adults, including age-friendly environments. Namely, older populations living in environments with age-friendliness will report high levels of active aging. Moreover, the factor associated with active aging included health behaviors that the older individuals with healthier lifestyles were more likely to have; they were more likely to have more active aging. Additional, factors related to active aging were educational level, age, and marital status. Namely, older adults with higher education, were younger, and had partners will perceive a higher level of active aging than their counterparts. Moreover, the older populations from Myanmar are less likely to rate themselves as active (see Table 4).

### 3.4. Age-Friendly Environments, Healthy Behaviors, and Socio-Demographic Factors as Predictors of Active Aging among ASEAN Older Populations

The results of multiple logistic regression in Table 5 indicated that predictors for active aging among the study sample include living in age-friendly environments, adopting healthy lifestyles, age level, marital status, and educational level, adjusted for the country of residents. Namely, older people living in a community with high levels of age-friendliness will be 5.52 times more active than those in a community with low levels of age-friendliness. Furthermore, older people with high levels of healthy lifestyles will be 4.93 times more active than those with lower levels of healthy lifestyles. Additionally, older adults with higher education, those aged between 55 and 64 years, and those with partners will be 2.61, 1.63, and 1.70 times more active than individuals with at least primary education, those aged 75 years or higher, and individuals that are separated, respectively.

### 3.5. Different Dimensions of Age-Friendly Environments as Predictors of Active Aging among ASEAN Older Population

Table 6 shows different age-friendly environments as predictors of ASEAN active aging. Adjusting for all related factors, we found that the positive age-friendly predictors are (1) a neighborhood with respect and social inclusion, (2) neighborhood with job support, (3) neighborhood with enough elderly parking lots, and (4) neighborhood with more accessible bus stations. Namely, the older individuals living in a neighborhood with more respect and social inclusion, more job support, enough elderly parking lots, and more accessible bus stations would be more active than their counterparts.

## 4. Discussion

Among a large sample (2031) of older adults in ASEAN member states, we found significant associations between active aging and perceived age-friendly environments, lifestyle behaviors, and socio-demographic factors. Overall, older people living in a community with high levels of age-friendliness will be 5.52 times more active in contrast to older people living in a community with low levels of age-friendliness. Furthermore, older people with high levels of healthy lifestyles will be 4.93 times more active than those with low levels of healthy lifestyles. Additionally, older adults with partners, those with education above high school education, and those aged between 55 and 64 years will be 1.70, 2.61, 1.63, and 1.70 times more active than those that are separated, with at least primary school education, and aged at 75 years or higher, respectively. Among age-friendly environments, the positive significant age-friendly predictors are a neighborhood with (1) respect and social inclusion, (2) job support, (3) enough elderly parking lots, and (4) more accessible bus stations. Namely, the older individuals living in a neighborhood with more respect and social inclusion, more job support, enough elderly parking lots, and more accessible bus stations would be more active than their counterparts.

This study found a significantly positive relationship between active aging and neighborhoods with respect and social inclusion, which supports previous studies [18,19,20]. Therefore, it reinforces the fact that living in a supportive social environment, such as a community with respect and social inclusion, can strengthen social capital, resulting in a more physically active senior, and this can result in more active aging. Furthermore, regarding transportation accessibility, we found that effective transportation services such as elderly parking lots and accessible bus stations are significant facilitators associated with older people’s active aging, which is a result that is similar to previous studies [15,16,17]. Therefore, improving transport service for the older population, for example, increasing accessible bus stations or providing more older parking lots, is significant for promoting the activities of the older population, including ASEAN older people. In addition, our research supports the significant role of a neighborhood with job support, especially for older people with economic deprivation. A previous study observed that the perception of active aging varied by culture and socioeconomic level [31]. For example, the older population with professional jobs in high-income countries such as European countries and America [31,32] perceived health and social functioning as essential parts of active aging. By contrast, lay older people, particularly those in low and middle-income countries of Asia, weighed the concept of active aging in other issues such as financial security, adequate income, sufficient pension, social security, and other benefits and discounted social services [31,33,34,35]. Therefore, it is challenging for the policy makers to achieve financial security associated with active aging with respect to the older population who encounter economic deficiency, whether in developed or developing countries [10,11,36].

Our result also strongly supports the positive role of healthy lifestyles on active aging, which is similar to the earlier research [21,22,23,24,25]. Therefore, we confirm that providing a healthy diet with high intakes of fruit and vegetables, legumes, grains and cereals, fish and seafood, and nuts can be an excellent choice to improve a senior’s health and to remain active. Furthermore, in addition to a healthy diet, we agree that physical activity is a good habit for seniors to maintain active aging. Some interesting physical activity for older adults include exercising the physical and mind through intellectual stimulation, inquiry, and curiosities that can sharpen older cognitive abilities [21,24]. Moreover, a multicomponent exercise incorporating physical, cognitive, and social activities, called “cognicise”, is good for maintaining physical health and cognitive ability associated with active aging [25,37]. Finally, we strongly support sports participation as a standard policy to help older people in ASEAN maintain their independence and to improve their general quality of life.

Regarding socio-demographics, as expected, younger age, higher education, and married/coupled are positive determinants for active aging. Indeed, education could assist in prolonging working life, which includes both paid and voluntary [38]. Therefore, highly educated older adults have more chances to extend their working life; they can be more energetic and active than the seniors without a job. Another part of practical education is lifelong learning. Previous research [34] claimed that learning could improve an individual’s skills, experiences, intelligence, memory, and perception. Therefore, continuing education as lifelong learning should be another policy for developing older persons’ well-being and quality of life [39]. With respect to marital status, our findings are in line with previous studies from Thailand, which revealed that family warmth from a spouse and children was positively related to the active aging of older adults in Thailand [40]. On the other hand, another finding [41] revealed that loneliness from losing a spouse through death experiences would increase poor mental well-being in older individuals. Therefore, alleviating loneliness associated with elderly loss and emphasizing the effort of the community and society to alleviate loneliness are challenges for promoting active aging among the older population.

The results of this study can assist planners in designing an age-friendly environment to promote active aging among the ASEAN older population. Firstly, it is reasonable to consider creating supportive social environments. That is, a community with respect for the elderly and social inclusion of the elderly can strengthen the social capital associated with active aging. Secondly, improving transportation services for the older population, such as increasing accessible bus stations or providing more parking spaces, is significant for built environments associated with raising the active aging of the ASEAN older adults. In addition, it is necessary to prolong employment for increasing financial security related to the well-being of the senior population. Finally, at the individual level, our results implied that maintaining a healthy lifestyle, engaging life-long education, and living with a partner are the determinants of active aging for the ASEAN older population. Therefore, to maintain active living in senior individuals, especially with the difficulty of the COVID-19 pandemic, we strongly support the significant roles of promoting healthy lifestyles, including maintaining a healthy diet and engaging in regular physical activity. Additionally, we also support the importance of lifelong learning, which mainly includes digital literacy. We believe that digital knowledge can facilitate senior people in updating helpful information, primarily precious health, and social services for everyday living. Furthermore, digital technology can alleviate loneliness for the seniors who are physically distancing.

There are some limitations to this study. First, the nature of the study’s cross-sectional design cannot confirm the causal relationships between active aging and the predictors. Thus, future research might include a larger population quantitative sample and use structural equation modeling to explain its direct and indirect effects. Second, we measured an age-friendly environment by using questionnaires with subjective judgment. Therefore, we recommend conducting future research using objective measures. Finally, another limitation of the study was that our sample size contained almost twice as many females than males. In this case, further investigation should study active aging based on gender perspectives so that decision-making can be relevantly addressed. However, the strength of the present study is that it is an innovative study that combines a sample of older adults in the ASEAN in an effort to promote active aging in this region. Therefore, it is a valuable study that can allow policymakers to understand the determinants of active aging in order to prepare for the appropriate aging community in the ASEAN region.

## 5. Conclusions

Significant positive predictors of active aging among the ASEAN population include perceived high level of age-friendly environments, healthy lifestyles, higher educational level, and living with a partner. Among age-friendly environment predictors, the positive significant age-friendly predictors are neighborhoods with (1) respect and social inclusion, (2) job support, (3) enough elderly parking lots, and (4) more accessible bus stations. Therefore, the proposed community-based age-friendly policies to promote active aging among the ASEAN older population include the following: firstly, creating supportive social environments such as a community with respect and social inclusion; secondly, improving transportation services for the older people such as increasing accessible bus stations or providing more older parking lots; and finally promoting economic security by prolonging employment for the senior population. Furthermore, at the individual level, our study indicated that older people with healthy lifestyles, higher educational levels, and living with partners would be more likely to be active than their counterparts. Therefore, we strongly support the importance of promoting healthy lifestyles with healthy diets and being physically active in everyday life in the older population. Additionally, we strongly support advanced lifelong learning, which mainly includes digital literacy, so that older people can update helpful information and alleviate the loneliness from physical distancing, especially in the current COVID-19 pandemic.

## Figures and Tables

**Table 1 ijerph-18-08290-t001:** Sample characteristics regarding socio-demographic and behavioral factors by country.

Variables	Country									
	Malaysia (*N* = 537)		Vietnam (*N* = 497)		Myanmar (*N* = 487)		Thailand (*N* = 510)		Total (*N* = 2031)	
	*N*	%	*N*	%	*N*	%	*N*	%	*N*	%
1. Gender										
1.1 Male	233	43.4	212	42.7	164	33.7	146	28.6	755	37.2
1.2 Female	304	56.6	285	57.3	323	66.3	364	71.4	1276	62.8
2. Education										
2.1 At least primary education (under 6 years of education)	19	3.5	165	33.2	415	85.2	313	61.4	912	44.9
2.2 Secondary education (6–9 years)	90	16.8	247	49.7	68	14.0	110	21.6	515	25.4
2.3 Higher education (more than 9 years)	428	79.7	85	17.1	4	0.8	87	17.1	604	29.7
3. Age level										
3.1 55–64 years old	376	70.0	185	37.2	201	41.3	200	39.2	962	47.4
3.2 65–74 years old	144	26.8	218	43.9	191	39.2	194	38.0	747	36.8
3.3 75 years old or more	17	3.2	94	18.9	95	19.5	116	22.7	322	15.9
4. Marital status										
4.1 Single	50	9.3	12	2.4	17	3.5	80	15.7	159	7.8
4.2 Married/coupled	414	77.1	375	75.5	230	47.2	211	41.4	1230	60.6
4.3 Separate/divorce/widow	73	13.6	110	22.1	240	49.3	219	42.9	642	31.6
5. Heathy lifestyles (Cut point = 14)										
5.1 Low	257	47.9	161	32.4	433	88.9	277	54.3	1128	55.5
5.2 High	280	52.1	336	67.6	54	11.1	233	45.7	903	44.5

**Table 2 ijerph-18-08290-t002:** Descriptive statistics of perceived age-friendly environments of ASEAN older populations.

Items	Low *		Medium *		High *	
	Number	Percent %	Number	Percent %	Number	Percent %
1. Neighborhood walkability	791	38.9	631	31.1	608	29.9
2. Neighborhood public spaces and building	976	48.1	558	27.5	497	24.5
3. Neighborhood with respect and social inclusion	517	25.5	629	31	885	43.6
4. Neighborhood job support	1356	66.8	342	16.8	333	16.4
5. Neighborhood health and services information	756	37.2	601	29.6	674	33.2
6. Neighborhood accessible bus station	1095	53.9	468	23	468	23
7. Neighborhood elderly parking lot	1124	55.3	472	23.2	435	21.4
8. Total neighborhood age-friendly environment (Min-Max: 0–28) Cut point: low (0–7) medium (8–15), high (16–28)	567	27.9	946	46.6	518	25.5

Note *: Each age-friendly environment is scored from 0 to 4 on a response ordinal scale (not at all, a little, moderately, mostly, and extremely). We categorized each item as low, medium, and high with 1 = low (not at all, a little), 2 = medium (moderately), and 3 = high (mostly, extremely).

**Table 3 ijerph-18-08290-t003:** Descriptive statistics of perceived active aging of ASEAN older populations.

Items	Level			
	Low		High	
	Number	Percent	Number	Percent
Self-Reliance (Min-Max: 8–32) Cut point:25	923	45.4	1108	54.6
Participation (Min-Max: 8–32) Cut point:18	1178	58.0	853	42.0
Life security (Min-Max: 10–40) Cut point:25	1041	51.3	990	48.7
Overall active aging (Min-Max: 28–103) Cut point:68	1024	50.4	1007	49.6

**Table 4 ijerph-18-08290-t004:** Factors associated with active aging using Chi-square statistics.

Variables	Level of Overall Active Aging			
	Low		High	
	Number	Percent	Number	Percent
1. Gender				
Male	363	48.10%	392	51.90%
Female	661	51.80%	615	48.20%
2. Educational level **				
2.1 At least primary education (under 6 years of education)	661	72.50%	251	27.50%
2.2 Secondary education (6–9 years)	196	38.10%	319	61.90%
2.3 Higher education (more than 9 years)	167	27.60%	437	72.40%
3. Age level **				
55–64 years old	411	42.70%	551	57.30%
65–74 years old	403	53.90%	344	46.10%
75 years old or more	210	65.20%	112	34.80%
4. Marital status **				
Single	73	45.90%	86	54.10%
Married/coupled	513	41.70%	717	58.30%
Separate/divorce/widow	438	68.20%	204	31.80%
5. Healthy lifestyle **				
Low	806	71.50%	322	28.50%
High	218	24.10%	685	75.90%
6. Perceived age-friendly neighborhood **				
Low	467	82.40%	100	17.60%
Medium	424	44.80%	522	55.20%
High	133	25.70%	385	74.30%
7. Country **				
Malaysia	169	31.50%	368	68.50%
Vietnam	162	32.60%	335	67.40%
Myanmar	446	91.60%	41	8.40%
Thailand	247	48.40%	263	51.60%

Note: Low active aging was defined as the total score of active aging equal to or below 68, whereas high active aging was defined as the total score of active aging above 68; ** *p* < 0.001.

**Table 5 ijerph-18-08290-t005:** Age-friendly environments, behavioral determinants, and socio-demographic factors as predictors of total active aging among the older adults.

Predictors	Adjusted Odds Ratio	95% CI	
		Lower	Upper
Gender			
Female	1.02	0.80	1.30
Male (Reference)			
Educational level			
Secondary education (6–9 years) **	2.06	1.53	2.77
Higher education (more than 9 years) **	2.61	1.83	3.73
At least Primary education(under 6 years of education) (Reference)			
Age level			
55–64 years old	1.63	1.15	2.33
65–74 years old	1.11	0.78	1.58
75 years old or more(Reference)			
Marital status			
Single	1.36	0.88	2.09
Married/coupled **	1.70	1.29	2.24
Separate/divorce/widow (Reference)			
Healthy behaviors			
High **	4.93	3.91	6.21
Low (Reference)			
Perceive age-friendly environment			
Medium **	2.23	1.61	3.09
High **	5.52	3.67	8.30
Low (Reference)			
Country			
Malaysia	0.86	0.60	1.24
Vietnam *	0.61	0.43	0.86
Myanmar **	0.22	0.15	0.34
Thai (Reference)			

CI = Confidence Interval; Adjusted Odds Ratio using “enter” with Hosmer and Lemeshow goodness of fit; Chi-square 12.07, df8, and 0.15; * *p* < 0.05; ** *p* < 0.01.

**Table 6 ijerph-18-08290-t006:** Different dimensions of age-friendly environments as predictors of active aging towards ASEAN older population.

Predictors	Adjusted Odds Ratio	95% CI	
		Lower	Upper
Neighborhood walkability			
Medium	1.16	0.81	1.67
High	0.93	0.60	1.45
Low (Reference)			
Neighborhood public spaces and building			
Medium	0.87	0.60	1.24
High	1.16	0.75	1.79
Low (Reference)			
Neighborhood with respect and social inclusion			
Medium **	2.53	1.69	3.79
High **	9.23	6.13	13.90
Low (Reference)			
Neighborhood available job support			
Medium **	1.87	1.34	2.62
High **	1.92	1.35	2.73
Low (Reference)			
Neighborhood with health and services information			
Medium	0.90	0.65	1.26
High	0.87	0.63	1.20
Low (Reference)			
Neighborhood accessible bus station			
Medium **	1.71	1.24	2.37
High	0.92	0.66	1.28
Low (Reference)			
Neighborhood elderly parking lot			
Medium *	1.56	1.12	2.17
High **	1.92	1.29	2.86
Low (Reference)			

CI = Confidence Interval; Adjusted Odds Ratio using “enter” with Hosmer and Lemeshow goodness of fit; Chi-square 5.92, df8, 0.66; * *p* < 0.05; ** *p* < 0.01; controlling for gender, age level, educational level, marital status, health behavior, and country of residence.

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
