# Peer review of "Active Aging in ASEAN Countries: Influences from Age-Friendly Environments, Lifestyles, and Socio-Demographic Factors"

_ijerph, 2021, doi:10.3390/ijerph18168290_

Round 1

Reviewer 1 Report

The purpose of the article stated by the authors (6 authors) is to examine the significant predictors of active aging among older adults in ASEAN. They took three key pillars of active aging suggested by the World Health Organization, including self-reliance and independence, participation, and maintaining life security. They share belief that the state of health in older age is influenced by  the determinants as age-friendly environments, behavioral determinants, and socio-demographic factors.

Areas of strength:

  • Extensive research – the sample consists of 2,031 older people aged 55 years and over, in Malaysia, Myanmar, Vietnam, and Thailand.
  • Clearly described methodology of research
  • Comparison of survey results on active aging for ASEAN's low and middle-income countries and selected results  of research from Western countries as the US, Canada

Areas of weakness:

  • there is no satisfactory answer as to whether the income level of older people somehow determines the similarity of attitudes towards active ageing when comparing low- and middle-income countries and countries with a higher standard of living (US, Canada, Hongkong) than the ASEAN countries surveyed.

Author Response

Dear Reviewer;

We are highly appreciated your kindly review. We already add the information as your recommendation in the manuscript as blue letter hereunder:

In addition, our research supports the significant role of a neighborhood with job support, especially for older people with economic deprivation. A previous study addressed that the perception of active aging varied by culture and socioeconomic level[31]. For example, the older population with professional jobs in high-income countries such as European countries and America[31,32] perceived health and social functioning as essential parts of active aging. In contrast, lay older people, in particular low and middle-income countries of Asia, weighed the concept of active aging in other issues such as financial security: adequate income, sufficient pension, social security, other benefits, and discounted social services[31,33-35]. Therefore, it is challenging for the policy makers to achieve financial security associated with active aging towards the older population who encounter economic deficiency, whether in developed or developing countries [10,11,36].

Thank you very much again for your valuable time.

Warmest Respectfully,

Sariyamon and team

Reviewer 2 Report

In the context of Healthy Aging, the present cross-sectional quantitative research reports a significant effort f(2031 older people, +55 years old) from four South East Asian countries, namely, Thailand, Malaysia, Vietnam, and Myanmar, to analyze perceived age-friendly environments, perceived active aging, behavioral determinants, and socio-demographic variables. The aim was to search for predictors of active aging. Factors associated with active aging were a healthier lifestyle, younger age, higher educational level, and partner. Age-friendly environment predictors were identified and will be helpful for public health policymakers. The work is well conducted and presented. However, due to the relevance of the work and the gender bias in most of the factors and variables analyzed, the data under a gender perspective should also be provided so that decision-making can address the expected gender bias in these predictive profiles.

Line 217 and somewhere else: Elderly

Please, note: The word 'elderly' is considered ageism. I strongly recommend using 'older adult' or 'older person' instead. (please, check, https://journals.lww.com/jgpt/fulltext/2011/10000/use_of_the_term__elderly_.1.aspx)

Minor

Lines 158 159 Perceive -- perceived

Among age-friendly environment predictors, the positive significant age-friendly 364 predictors are 1) neighborhood with respect and social inclusion,2) neighborhood with 365 job support,3) neighborhood with enough elderly parking lots, and 4) neighborhood with 366 more accessible bus stations.

Please, simplify so that the factors are the focus. For instance:
The positive significant age-friendly predictors are a neighborhood with 1) respect and social inclusion, 2) job support, 3) enough elderly parking lots, and 4) more accessible bus stations.

Author Response

Dear Reviewer;

We are highly appreciated your kindly review. We already fixed the manuscript per your recommendations; please consider the yellow letter.

Thank you very much again for your valuable time.

Warmest Respectfully,

Sariyamon and team

Reviewer 3 Report

Dear authors,

Please find attached a pdf file including comments for the proposed article.

Also check the article Published: 23 June 2020 entitled:  

Age-Friendly Environments in ASEAN Plus Three:
Case Studies from Japan, Malaysia, Myanmar,
Vietnam, and Thailand. 

We can check that the proposed article is extracted from the same study using the same data only excluding the sample from Japan.

In fact the recommendations and guidelines for the study are in great controversy with the updates governments and WHO recommendations  since 2020.

We strongly believe that the proposed article is irrelevant and more constructive efforts should be done using relevant up to date data. 

All along the article we were trying in vain to understand the time frame of the study: when, and the ways the interviews were conducted.

Best Regards

Author Response

Dear Reviewer;

We are highly appreciated your kindly review.

  • As you noticed, we collected the data in five countries: Japan, Malaysia, Vietnam, Thailand, and Myanmar. Unfortunately, we can only collect preliminary data of age-friendly environments from Japan due to resource limitations. Nevertheless, we accomplished more data collection for the other four countries and have different perspectives to consider. Therefore, in this article, we focus on active aging that was not mentioned in the previous article that you referred to. Then, please kindly reconsider this comment.
  • The survey was conducted during the three months of January to March 2019, and we had already added this information to the manuscript.
  • We already fixed the manuscript per your recommendations; please consider the red letter.

Thank you very much again for your valuable time.

Warmest Respectfully,

Authors

Round 2

Reviewer 2 Report

The authors have properly addressed the questions raised and implemented the corrections ('elderly') and suggestions (to be more clear in the conclusions, to add the gender perspective in the discussion). 
It's a pleasure to endorse they work,

Author Response

Dear reviewer,

Thank you very much for your kindly action. We are highly appreciated your valuable time to consider our manuscript.

Warmest Respectfully,

The authors

Reviewer 3 Report

Good morning authors, I thank you for your reply and effort spent in executing your current paper.  Attached the link to google drive file with green highlights and comments. Trying to clear my second review here are the thoughts and reflections over your study research. Line 2: reconsider the titIe: Impact of Age-Friendly Environment, Lifestyle and, Socio-Demographic Factors on Active Ageing. Line 15: The concept of Active Aging is defined as “…the process of optimizing opportunities for health, participation and security in order to enhance quality of life as people age” Ref:  WHO. Active Aging: A Policy Framework. Geneva: World Health Organization (2002). Line 15: Healthy ageing is the focus of WHO’s work on ageing between 2015 – 2030. Healthy ageing replaces the World Health Organization’s previous focus on active ageing, a policy framework developed in 2002. Healthy ageing, like active ageing, emphasizes the need for action across multiple sectors and enabling older people to remain a resource to their families, communities and economies. Ref:https://www.who.int/westernpacific/news/q-a-detail/ageing-healthy-ageing-and-functional-ability  Line 23: remove behaviors L25: please elucidate the meaning of active aging cities and age friendliness community  L 26: ?? unclear.  L 36: The active aging model as presented by the World Health Organization (WHO) encompasses six groups of determinants, each one including several features: 
(1) availability and use of health and social services (e.g., health promotion and prevention; continuous care); 
(2) behavioral determinants (e.g., exercise and physical activity; drinking and smoking habits; feeding; medication); (3) personal determinants (biology and genetics, and psychological characteristics); 
(4) physical environment (e.g., safety houses, low pollution levels); (5) social determinants (e.g., education, social care), and  (6) economic determinants (e.g., wage, social security). This group is complemented by two crosscutting determinants—gender and culture. According to this model, the key elements of active aging are (1) autonomy, which is the perceived ability to control, cope with, and make decisions about how one lives on a day-to-day basis, according to personal rules and preferences; (2) independence, which refers to the ability one has to perform functions related to daily living, i.e., the capability of living in the community with no and/or little help from others; (3) quality of life; and (4) healthy life expectancy, which refers to how long people can expect to live in the absence of disabilities. The main pillars of the model are participation, health, and security. More recently, a fourth pillar has been added to the model: lifelong learning.
Ref:  ILC-Brazil. Active Ageing: A Policy Framework in Response to the Longevity Revolution. Rio de Janeiro: ILC (2015). L36:   In terms of regions, over half of the world’s older people live in Asia. Asia’s share of the world’s oldest people will continue to increase the most while Europe’s share as a proportion of the global older population will decrease the most over the next two decades. L39: An active ageing approach to policy and 
programme development has the potential to  address many of the challenges of both individual and population ageing. L41: The Determinants of Active Ageing: While Active ageing depends on individual, and national influences or “determinants”. Understanding the evidence we have about these determinants helps us design policies. What we know about how the broad determinants of health affect the process of ageing needs clarify and specify the role of each determinant, as well as the interaction between determinants, in the active ageing process. We also need to better understand the pathways that explain how these broad determinants 
actually affect health and well being. (WHOref_initiative). L51:Please check the  following: 1-Gender and 2- Cultural determinants along with the 6 others determinants specified at the WHO reference.
Determinants Related to Health and Social Service Systems:1- Health Promotion and Disease Prevention
2- Curative Services3- Long-term care 4- Mental Health Services 5- Tobacco Use 6- Physical Activity 7-Healthy Eating 8- Oral Health 9- Alcohol. 10- Access to medication. 11- Adherance. Personal factors: (genetical, psychological, bological).  Envionmental: Clean water, clean air and access to safe foods ... Social Factors: Economical Factors: income, social protection and work. L106: Determine the determinants that is used in the study as a relevant input for the study analysis and discussion. L108: Aim of the study: Aim of the study: The study aims to elucidate ... develop, test, determinants of age friendly ... Please specify and state the aim of the study. The effect of the three chosen determinants of age friendly criteria as a parameter of active ageing. ??? Please elaborate. L210: Table1: Total 100% ... no illiterate in the sample?  where they excluded from the sample???  Thailand literacy rate for 2018 was 93.77%, a 0.9% increase from 2015. According to UNESCO Burma - Myanmar it has an adult literacy rate of 75.55%. Vietnam literacy rate for 2018 was 95.00% where the illiterate added to the (at least primary education)? L284: discussion on the bases of Perception of the three determinants as a element to be able to measure friendly ageing.  Reliability of the study. L373: The conclusion id too vage and are not extracted from the mentioned data. 
Seemingly to be able to publish such an article it is important to mention the post coronavirus WHO recommentations ... To what extent can policy makers adhere to such a controversial situation. 
The readers of an international journal of public health and environment will sure be more interested in reading something that goes on line with the dissonance in policy makers and WHO recommendations when it comes to friendly active ageing. L386: Is the conclusion too vague?  and are not extracted from the mentioned data? 
Seemingly to be able to publish such an article it is important to mention the post coronavirus WHO recommentations ... To what extent can policy makers adhere to such a controversial situation. 
The readers of an international journal of public health and environment will sure be more interested in reading something that goes on line with the dissonance in policy makers and WHO recommendations when it comes to friendly active ageing. L463: the reference online has evolved : https://extranet.who.int/kobe_centre/en?ua=1 
With my best regards,

Author Response

Dear Reviewer

Thank you very much for your valuable time to make the comments for our manuscript.

We try to fix the manuscript based on your recommendations. Please consider the blue items on the revised manuscript:

  • We fix the title but keep the first word as “Active aging” for our preference.
  • We cut behavior out on the line 23
  • For easier understanding, on the line 26 , we changed the age-friendliness community into age-friendly environments.
  • We add the detail of education by specific numbers of academic years on tables 1, 4, and 5
  • For the discussion part, we add information regarding the covid-19 prevention, consider the blue letter
  • For the conclusion, we rewrite this part based on the fixed discussion.

We hope our revision will satisfy you well.

Thank you again for your valuable comments,

The authors
